# Stress Reduction Potential in Mice Ingesting DNA from Salmon Milt

**DOI:** 10.3390/biology12070978

**Published:** 2023-07-10

**Authors:** Keiko Unno, Kyoko Taguchi, Mica Fujita, Keita Sutoh, Yoriyuki Nakamura

**Affiliations:** 1Tea Science Center, University of Shizuoka, 52-1 Yada, Suruga-ku, Shizuoka 422-8526, Japan; gp1719@u-shizuoka-ken.ac.jp (K.T.); yori.naka222@u-shizuoka-ken.ac.jp (Y.N.); 2Fordays Co., Ltd., Koami-cho, Nihonbashi, Chuo-ku, Tokyo 103-0016, Japan; m.fujita@fordays.jp (M.F.); k.sutoh@fordays.jp (K.S.); 3United Graduate School of Agricultural Science, Tokyo University of Agriculture and Technology, 3-5-8 Saiwai-cho, Fuchu-shi, Tokyo 183-8509, Japan

**Keywords:** adrenal hypertrophy, confrontational stress, cathepsin B, crowding stress, DNA, IL-1β, inflammation

## Abstract

**Simple Summary:**

This study was conducted to elucidate the functionality of food-derived nucleotide supplementation. DNA sodium salt derived from salmon milt (DNA-Na) was found to exhibit stress-relieving effects in a screening experiment using a stress load based on the territoriality of male mice. On the other hand, RNA from yeast showed no significant effect. The stress-relieving effect of DNA-Na was then examined in another experimental system, whereby mice were subjected to chronic crowding stress as they aged. We found that DNA-Na suppresses the inflammatory response in the hippocampus of aged mice. This suggests that the intake of dietary DNA suppresses stress-induced inflammation in the brain, which increases with aging.

**Abstract:**

The functionality of food-derived nucleotides is revealed when nucleotide components are ingested in emergency situations, such as during stress loading, though it is difficult to elucidate the physiological function of dietary nucleotide supplementation. Using a stress load experimental system utilizing territoriality among male mice, we evaluated whether DNA sodium salt derived from salmon milt (DNA-Na) has stress-relieving effects. It was found that stress was reduced in mice fed a diet containing a 1% concentration of DNA-Na, but this was insignificant for yeast-derived RNA. Next, we attempted to elucidate the anti-stress effects of DNA-Na using another experimental system, in which mice were subjected to chronic crowding stress associated with aging: six mice in a cage were kept until they were 7 months of age, resulting in overcrowding. We compared these older mice with 2-month-old mice that were kept in groups for only one month. The results show that the expression of genes associated with hippocampal inflammation was increased in the older mice, whereas the expression of these genes was suppressed in the DNA-Na-fed group. This suggests that dietary DNA intake may suppress inflammation in the brain caused by stress, which increases with age.

## 1. Introduction

Abundant in a variety of foods, nucleic acids are ingested and used to synthesize polynucleotides such as DNA and RNA through the salvage pathway. They are generally thought to be absorbed through the gastrointestinal tract in the form of nucleosides [1,2]. Although the quantity of nucleic acids of a dietary origin that is absorbed as nucleosides is not yet known, the majority of nucleosides and nucleobases primarily absorbed in the upper small intestine are immediately degraded in the small intestinal epithelial cells. They then pass through the portal vein to the liver, before entering blood circulation and being utilized by various organs. Nucleotides are a group of compounds that do not provide direct energy sources or biological components, but work to maintain homeostasis in the body, such as immunity, antioxidant function, and signal transduction, and are known to have various physiological functions [3,4,5]. Since nucleotides are also synthesized in vivo, it is difficult to elucidate the functionality of food-derived nucleotides when supplemented. However, the functionality of nucleotide supplementation within the body’s stress response may be clarified by ingesting nucleotide components during stress loads and other emergencies [6,7]. It has also been reported that, in the brain, synthesis in the salvage pathway predominantly occurs via de novo synthesis [8].

On the other hand, cell-free DNA fragments in the plasma/serum, which were released from dying cells, have been reported to induce stress responses [9,10]. In addition, stress loading increases cell-free DNA in the plasma [11]. The extracellular DNA of nuclear and mitochondrial origin, which used to be a passive marker of dead cells, has recently come to be considered as a signaling molecule. Cell-free and GC-rich unoxidized DNA has been reported to trigger reactions with inflammatory components [12]. Then, when new nucleic acid synthesis is required as a stress response of the organism, the diet-derived DNA is used in the salvage pathway.

While moderate stress is necessary and has positive effects, excessive and/or prolonged stress can have significant physical and mental effects, including the development or worsening of depression, mood disorders, cardiovascular diseases, and aging-related diseases [13,14,15,16]. It has been shown that theanine, an amino acid unique to tea leaves, exhibits excellent anti-stress effects [17]. Establishing other food components with stress-relieving effects and their mechanisms of action can allow for dietary intervention to contribute to the maintenance of people’s mental and physical health.

Among the biological responses triggered by stressors, endocrine responses via the hypothalamus–pituitary–adrenal (HPA) axis are well known [18,19]. The stress-induced secretion of corticotropin-releasing hormone (CRH) from the hypothalamus stimulates the secretion of the adrenocorticotropic hormone (ACTH) from the anterior pituitary gland, which in turn stimulates the adrenal cortex and promotes glucocorticoid release. These are regulated by negative feedback mechanisms, but an excessive activation of the HPA axis causes adrenal hypertrophy [20]. Therefore, the anti-stress action of the target substance can be evaluated by comparing the degree of adrenal hypertrophy inhibition under stress-loading conditions. In the present study, the anti-stress effect of nucleotides was evaluated using a model of psychosocial stress evoked by short-term confrontational housing. That is, two male mice were housed in the same cage separated by a partition to establish a territorial imperative. Then, the partition was removed, and the mice were confrontationally co-housed. As a marker for stress response, changes in the adrenal glands were studied and compared with those of group-housed control mice. Significant adrenal hypertrophy, observed in mice under confrontational housing, had developed in 24 h and persisted for at least 1 week. Therefore, adrenal hypertrophy was suitable for evaluating the anti-stress effect [21].

In this study, we examined whether the ingestion of DNA sodium salt (DNA-Na) from salmon milt and RNA from torula yeast may prevent or alleviate psychosocial stress. This screening test revealed that DNA-Na has anti-stress effects, but RNA has few effects. To determine whether DNA-Na can reduce stress, we used another stress-loading model that we constructed. In this experimental system, fast-growing mice were kept in groups, resulting in overcrowding during the 6-month rearing period and a chronic stress load as they aged [22]. This experimental system was deemed appropriate considering that the effects of aging on the stress response are important, with stress enhancement being observed in older mice. On the other hand, long-term confrontation rearing is not a stress-loading condition, because adrenal hypertrophy is no longer significant after 10 or more days in stress-tolerant ddY mice [21]. In other words, acclimation occurs between the two mice, and there is little confrontation stress or overcrowding stress. Thus, we examined whether DNA-Na is effective in alleviating stress in aged mice under group-rearing conditions. To clarify the physiological function of nucleotides under stressful conditions, we investigated the stress response in vivo and focused on changes in the brain. In an attempt to elucidate the mechanism of action, we examined changes in gene expression in response to stress and inflammation in the hippocampus of mice that ingested DNA-Na.

## 2. Materials and Methods

### 2.1. Animals

An outbred strain (ddY) of four-week-old mice was purchased from Japan SLC Co. Ltd. (Shizuoka, Japan). The mice were kept under a 12 h light/dark cycle (8:00–20:00 light period) at room temperature (23 ± 1 °C) and 55 ± 5% humidity. Feed (AIN-93M, Clea Co., Ltd., Tokyo, Japan) and drinking water were provided ad libitum. Animal care and experiments were conducted in accordance with the guidelines for animal experiments at the University of Shizuoka. All experimental protocols were approved by the University of Shizuoka Laboratory Animal Care Advisory Committee (approval no. 195241, 13 June 2019, and approval no. 225354, 3 March 2022), and were in accordance with the guidelines of the US National Institutes of Health for the care and use of laboratory animals.

### 2.2. Reagents

DNA sodium salt (DNA-Na, Fordays Co., Ltd., Tokyo, Japan) was prepared from salmon milt; HPLC analysis showed that the deoxyribonucleotide content in DNA-Na was 77.86% (*w*/*w*): 20.02% dAMP, 22.53% dTMP, 20.11% dGMP, 14.93% dCMP, and 0.26% deoxyribonucleoside. RNA was purified and concentrated from torula yeast. The RNA concentration was 73.68% (*w*/*w*): 18.53% AMP, 16.00% UMP, 20.67% GMP, 14.66% CMP, and 3.82% ribonucleoside.

### 2.3. Screening Method for Anti-Stress Effects

Four-week-old mice were housed in a group of six in a cage for 5 days, for acclimation. The mice were then divided into two groups: confrontationally housed and group-housed. For confrontational housing, a standard polycarbonate cage was divided into two identical subunits by a stainless-steel partition, as previously described [21]. As shown in Figure 1, two mice were housed in a partitioned cage for 1 week (single housing); the partition was then removed to expose the mice to confrontational stress for 24 h (confrontational housing). A dominant–inferior relationship was quickly established between the two mice, but there was no violent struggle that led to the death of either one. It was confirmed that there was no significant difference in the degree of adrenal hypertrophy between the dominant and recessive mice [21]. This suggests that the two mice under confrontational rearing were almost equally stressed. Adrenal hypertrophy was maximal one day after the start of confrontation rearing, and in the case of the ddY mice, this hypertrophy persisted up to 7 days; however, after 10 days, the adrenal enlargement declined [21].

Food containing each concentration of DNA-Na or RNA in AIN-93M (Clea Co., Ltd. Tokyo, Japan) was fed from the commencement of single housing. It has been found that the effects of anti-stress substances during stress loading are more effective when ingested before stress loading than when ingested after stress loading [21]. Therefore, in this study, DNA-Na was also ingested from the beginning of stand-alone rearing.

Group-housed mice were kept in groups of four or six mice per cage and fed the same diet containing each nucleotide. Each cage was placed in a Styrofoam box in order to prevent social contact between cages. Control mice were fed AIN-93M. The mice were anesthetized with isoflurane, and the adrenal and thymus glands were carefully dissected and wet-weighed.

### 2.4. Anti-Stress Effects of DNA-Na on Chronic Crowding Stress in Older Mice

The anti-stress effect was observed in mice that had ingested food containing 1% DNA-Na for one and six months. One-month-old mice were used at the start of this experiment. As shown in Figure 2, group-reared mice were kept in groups of six per cage for one and six months (group housing) [22]. The mice were 2 and 7 months old after each feeding period. During long-term rearing experiments, the group rearing of fast-growing mice is a condition under which overcrowding stress is produced.

The conditions of the confrontationally housed mice were as follows: two mice were housed in a partitioned cage for one week (single housing); the partition was then removed to expose the mice to confrontational stress for one and six months (confrontational housing). Stress from confrontational rearing has previously been observed to be lowered as a result of habituation between the two mice, as significant adrenal hypertrophy was no longer observed after more than 10 days [21]. In addition, confrontational rearing also caused brain atrophy in mice, but in the case of ddY mice, there was a subsequent recovery from atrophy [23]. Thus, after 6 months under confrontational rearing conditions, the mice were expected to be nearly stress-free, including from overcrowding stress.

The mice were fed food containing 1% DNA-Na in AIN-93M (Oriental Yeast Co., Ltd. Tokyo, Japan) from the commencement of single or group housing. The control mice were fed AIN-93M. Each cage was placed in a Styrofoam box in order to prevent social contact between cages.

Isoflurane was used to anesthetize the mice. The autopsy was performed between 10:00 AM and 12:00 AM. Blood was drawn from the jugular vein and corticosterone was measured from the serum obtained, using a commercially available ELISA kit (no. 501320, Cayman, MI, USA), following the manufacturer’s protocol. The hippocampus and cerebral cortex removed from the brain of each mouse were immediately frozen.

### 2.5. Quantitative Real-Time Reverse-Transcription PCR (qRT-PCR)

The hippocampi of mice were used for this analysis. Total RNA was isolated from a homogenized sample using a purification kit (NucleoSpin^®^ RNA, 740955, TaKaRa Bio Inc., Shiga, Japan) according to the manufacturer’s protocol. The resulting RNA was processed into cDNA using the PrimeScript^®^ RT Master Mix kit (RR036A, Takara Bio Inc.). qRT-PCR analysis was performed using the PowerUp™ SYBR™ Green Master Mix (A25742, Applied Biosystems Japan Ltd., Tokyo, Japan) and automated sequence detection systems (StepOne, Applied Biosystems Japan Ltd.). Relative gene expression was measured using previously validated primers for the neuronal PAS domain protein 4 (Npas4) [24], interleukin 1β (IL-1β) [25], tumor necrosis factor α (TNFα) [26], and cathepsin B (CtsB) [27] genes (Table 1). Furthermore, cDNA derived from transcripts encoding β-actin was used as the internal control.

### 2.6. Measurement of Oxidative Damage in the Brain and Serum

Malondialdehyde (MDA) is one of the lipid peroxidation breakdown products that is widely used as a major marker of lipid peroxidation. The content of MDA in the homogenate of the cerebral cortex was determined with a kit (MDA assay kit, Nikken Seil Co. Ltd., Fukuroi, Shizuoka, Japan) according to the protocol provided by the manufacturer. A brain sample was homogenized with the assay buffer in the kit. The homogenate obtained was slightly suspended, suggesting that the sample contained hemoglobin. According to the method described by Badcocka et al. [28], the hemoglobin was removed as follows. To 750 μL of the reaction solution, 500 μL of n-butanol was added, and after stirring, it was centrifuged at 10,000× *g* for 2–3 min. Of the 1 N NaOH solution, 500 µL was added to 400 μL of the butanol fraction obtained. After stirring well, 400 μL of the NaOH fraction was collected. Immediately, 100 μL of 3.7 M phosphoric acid was added. The absorbance spectrum of the reaction solution (200 μL) was measured from 400 to 700 nm.

The reactive-oxygen-metabolite-derived compounds (d-ROMs) and biological antioxidant potential (BAP) tests were used to evaluate the oxidated stress levels and antioxidant levels in blood serum. The d-ROM test can evaluate the amount of free radicals by measuring hydroperoxide in the blood, which is produced by free radicals and reactive oxygen species in the body. The BAP test evaluates the reducing power of blood in terms of its antioxidant capacity. These tests were performed using the free radical analyzer, FREE (Health & Diagnostics Limited Co., Parma, Italy), and the reagents for BAP and d-ROM were purchased from Wismerll Co., Ltd. (Tokyo, Japan). These reagents were used according to the manufacturer’s instructions.

### 2.7. Statistical Analyses

The results are expressed as the mean ± standard error. Statistical analysis was performed using a one-way ANOVA, and statistical significance was set at *p* < 0.05. Confidence intervals and the significance of differences in the means were estimated using Tukey’s honest significant difference method or Fisher’s least significant difference test.

## 3. Results

### 3.1. Suppression of Adrenal Hypertrophy by Ingestion of DNA-Na from Salmon Milt

In ddY mice, the anti-stress effects of DNA-Na (1% and 1.5%) and RNA (1% and 1.5%) as nucleotide products during single and confrontational housing were evaluated for their inhibitory effects on adrenal hypertrophy. This experiment was conducted twice. The results show that adrenal hypertrophy was significantly suppressed even under stress-loading conditions in mice fed a diet containing 1% DNA-Na, and tended to be lower, but not statistically significant, in mice fed 1.5% DNA-Na (Figure 3A, *p* = 0.0025). However, RNA had no significant effect.

The observation of effects on the thymus showed no significant thymic atrophy in ddY mice that had ingested DNA-Na or RNA (Figure 3B, *p* = 0.16).

### 3.2. Effects of DNA-Na Ingestion on Chronic Crowding Stress in Older Mice

Since it was found that DNA-Na has a stress-reducing ability, the effects of DNA-Na on long-term crowding stress were examined in mice. The ddY mice grew quickly, with their body weight increasing an average of 1.5-fold during the one-month rearing period. When rearing was continued under these conditions for 6 months, the body weight increased an average of 2.0-fold during the 6-month rearing period. Therefore, under the condition of six mice per cage (30 cm × 20 cm), crowding stress was considered to occur with aging. No fighting was observed between group-housed mice during this period. There was no significant difference in the body weights between the group and confrontational rearing.

The adrenal glands were smaller in older control group-rearing mice than in the younger ones (Figure 4A, *p* = 0.0003). The thymus gland is an age-sensitive organ and usually shows atrophy in older mice; thereby, no effect of DNA-Na ingestion was observed (Figure 4B, *p* = 0.0032). There was no significant difference in the adrenal and thymus weights between group- and conformational-reared mice. For confirming the wet weights of the adrenal glands and thymus, an additional experiment was performed on mice that were group-reared with six mice for 1 month, so the number of individuals resulted in 12 samples. For the others, the number of samples was six.

### 3.3. Effects of DNA-Na Intake on Serum Corticosterone Levels

Although hair loss was observed in the long-term group of control mice, no epilation was observed in the DNA-Na-fed mice of the same age group. Recently, corticosterone has been shown to inhibit follicular GAS6, a gene that encodes the secreted factor growth arrest specific 6 to govern hair follicle stem cell quiescence [29]. To estimate the degree of stress, changes in serum corticosterone levels were measured in mice that had been reared in group and confrontational housing. Corticosterone levels have a diurnal rhythm. In mice, corticosterone levels are usually low in the morning, but the rhythm and secretion change when stress is applied [21]. Therefore, we cannot simply compare group rearing to confrontational rearing, even at the same age. However, it is possible to compare the effects of DNA-Na diet intake on mice under the same rearing conditions at the same age. There were individual differences in corticosterone levels and no statistically significant differences were found at 1 and 6 months of age (Figure 5, *p* = 0.61).

### 3.4. Effects of DNA-Na Intake on Gene Expression in the Brain

Changes in gene expression associated with stress and inflammation were examined for the hippocampus via real-time PCR. The lysosomal enzyme, CtsB, which is involved in the induction of proinflammatory responses, was examined. The expression was significantly higher in older group-reared mice that ingested the control diet than in those that ingested DNA-Na (Figure 6, *p* = 0.014). Similarly, the expression of proinflammatory cytokine IL-1β was significantly increased in older group-reared mice fed the control diet (*p* = 0.020). However, the expression was significantly suppressed in older group-reared mice fed DNA-Na. On the other hand, the expression of another proinflammatory cytokine, TNFα, was not changed (*p* = 0.56). The expression of Npas4, a factor important for stress tolerance, was not significantly increased (*p* = 0.44).

### 3.5. Effects of DNA-Na Intake on Oxidative Damage

The effects of DNA-Na intake on oxidative stress were examined in the cerebral cortex and serum. MDA levels in the cortex were very low and did not differ significantly between the groups (Figure 7). The amount of serum hydroperoxides, as assessed via d-ROM, was not statistically different from that of the older mice, and the serum antioxidant activity, assessed via BAP, was significantly lower in the older mice. However, there were no differences between the groups with and without DNA-Na intake.

## 4. Discussion

To clarify the functionality of nucleotides in vivo, this study investigated whether nucleotide ingestion has a mitigating effect during stress loading. Short-term confrontational stress-loaded mice fed a diet containing 1% DNA derived from salmon milt (DNA-Na) showed a clear inhibition of adrenal hypertrophy, suggesting the stress-reducing effect of DNA-Na. However, mice subjected to long-term overcrowding stress did not show adrenal hypertrophy, changes in stress-responsive *Npas4* expression, oxidative injury, or changes in corticosterone levels. These may be the result of a hormesis phenomenon, in which short-term low-dose stress induces a stress response, but long-term high-dose stress has an opposite inhibitory effect [30,31]. Since various factors are involved in this hormesis, differences in feed may have an effect. It has been reported that the stress response is stronger with refined feed than with non-purified feed [32]. In fact, we observed significant adrenal hypertrophy and an increased expression of *Npas4* in long-term group-reared ddY mice that were fed a non-purified diet [22]. Purified feed was used with the expectation of a clear stress response, but the results were somewhat complicated. The absence of a dose dependence between 1% and 1.5% DNA-Na may also be explained by the hormesis phenomenon.

Next, we considered the difference between DNA and RNA actions. In the evaluation of adrenal hypertrophy based on the territoriality between male mice under the short-term confrontational rearing, we found that DNA-Na showed significant stress-relieving effects. However, RNA derived from yeast had no significant effect. The reason for the difference in the effects of DNA and RNA on stress is very interesting. For example, most nucleic acids in breast milk are ribonucleotides or RNA, which have been shown to be important nutrients for infants. In addition, the functionality of microRNAs in milk may be an important function of RNA [33]. On the other hand, circulating DNA fragments derived from the nuclei and mitochondria of dying cells have been shown to act as stress signaling molecules [9,10]. Since the stress load has been reported to increase DNA fragments in plasma [11], it is likely that endogenous cell-free DNA fragments increased in mice under the conditions of crowding stress. Among the cell-free DNA circulating in the body, DNA fragments with a high GC content and unoxidized DNA have been reported to cause weak, but long-lasting, reactions with inflammatory components [12]. The expression of inflammation-related genes, *CtsB* and *IL-1β*, was increased in mice subjected to prolonged overcrowding stress, suggesting that such GC-rich, unoxidized DNA fragments may have been elevated in the plasma of long-term stress-loaded mice. Dietary-derived DNA may have been utilized via the salvage pathway, resulting in suppression of inflammation-related gene expression. It has been reported that DNA oligonucleotides act differently depending on their length [34,35]. In the future, it will be necessary to examine the length, GC content, and degree of oxidation of DNA fragments in the plasma of mice that have been subjected to stress.

The increase in the leakage of CtsB in microglia with aging is involved in the increased inflammatory mediators [36], complicating the maintenance of a lysosomal acidic pH, which induces inflammatory signals [36]. Furthermore, increased CtsB has also been reported to increase IL-1β secretion [37,38,39,40,41]. The increase in inflammatory response in the hippocampus with aging suggests that the overcrowding stress accelerates brain aging, and that this is suppressed by DNA-Na. The *CtsB* expression was increased by DNA-Na ingestion in mice within the 1-month confrontation-reared group. The *TNFα* expression was unchanged. The reasons for these findings remain to be determined.

The dietary supplement of nucleotides is important during periods of rapid growth and stress [42,43]. Under stressed conditions, nucleic acid components such as dietary nucleosides supplied to the brain via the bloodstream may have been rapidly utilized in the stress response via the salvage pathway, resulting in the suppression of the inflammatory response.

Various methods of examining the effects of short- and long-term social stress have been reported [44]. Stress from confrontational rearing has previously been observed to be lowered as a result of habituation between the two mice, as significant adrenal hypertrophy was no longer observed after 10 days or more [21]. Thus, after 6 months under confrontational rearing conditions, the mice were expected to be nearly stress-free, including from overcrowding stress. On the other hand, the method of applying overcrowding stress through group-rearing, associated with weight gain, is a simple and interesting stress-loading system, allowing for the examination of the effects of aging on stress. Inflammatory responses and oxidative stress increase with aging [45]. Therefore, it is important to search for stress-relieving substances that take into account the effects of aging, and the experimental system of aging-related stress loading is considered to be highly valuable for use as an anti-stress evaluation system.

DNA-Na has been reported to improve brain function in mice [46,47]. A study of gene expression in the hippocampus of the brain showed an increased gene expression of markers for neurons, microglia, astrocytes, and oligodendrocytes, indicating that DNA ingestion may be involved in promoting differentiation into the various cells of the brain. In addition, increases in cytidine, deoxycytidine, and cytosine were observed in the hippocampus of the brain. This suggests that the increase in nucleic acids containing cytosine bases in the hippocampus is involved in the improvement of memory learning ability [47].

In addition, it has been shown that DNA-Na is a source of NAD as a nucleoside in alcohol metabolism in humans [48]. Moreover, reactive oxygen species generated during alcohol degradation are thought to be a major cause of liver injury in ethanol-induced liver injury, but the fact that lipid peroxidation was suppressed in model rats after the ingestion of salmon milt extract indicates that salmon milt extract has the ability to enhance antioxidant activity [49]. As a mechanism of action, experiments using hepatocytes have shown that deoxyadenosine in salmon milt extract may suppress liver inflammation [49]. On the other hand, salmon sperm DNA has also been reported to prevent liver inflammation [50,51].

DNA-Na supplementation significantly suppressed increases in *CtsB* and *IL-1β* mRNA associated with stress-induced hippocampal inflammatory responses. Long-term, low-grade inflammation is associated with aging. Since inflammation in the brain is associated with decreased brain function, the ability to reduce or suppress age-related inflammation in the brain via diet can help prevent brain aging [52]. For example, curcumin, sulforaphane, vitamin C, emodin, ellagic acid, resveratrol, epigallocatechin gallate, anthracyclines, short-chain fatty acids, pro- and pre-biotics, and caloric restriction have been reported to modify aging-related inflammation [53]. Similarly, DNA-Na may be an anti-inflammatory ingredient derived from a natural food component that inhibits brain inflammation, which is increased by stress and aging. It might be advisable to take DNA-Na supplements immediately after or before the start of an anticipated period of stress (surgery, enhanced loads, viral infection). Further studies are needed to elucidate the detailed mechanisms of how DNA ingestion produces anti-inflammatory effects.

## 5. Conclusions

The ingestion of salmon milt-derived DNA suppressed the expression of hippocampal inflammation-related genes *CtsB* and *IL-1β* mRNA levels induced by crowding stress.

## Figures and Tables

**Figure 1 biology-12-00978-f001:**
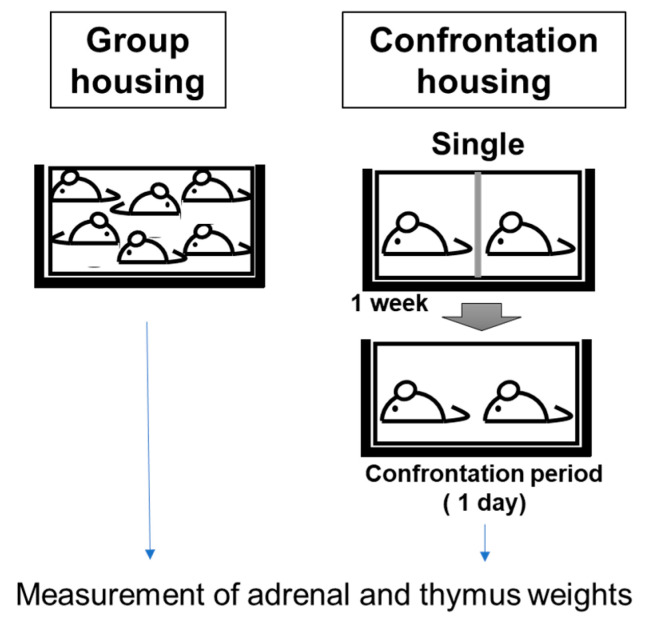
Screening method for anti-stress effects.

**Figure 2 biology-12-00978-f002:**
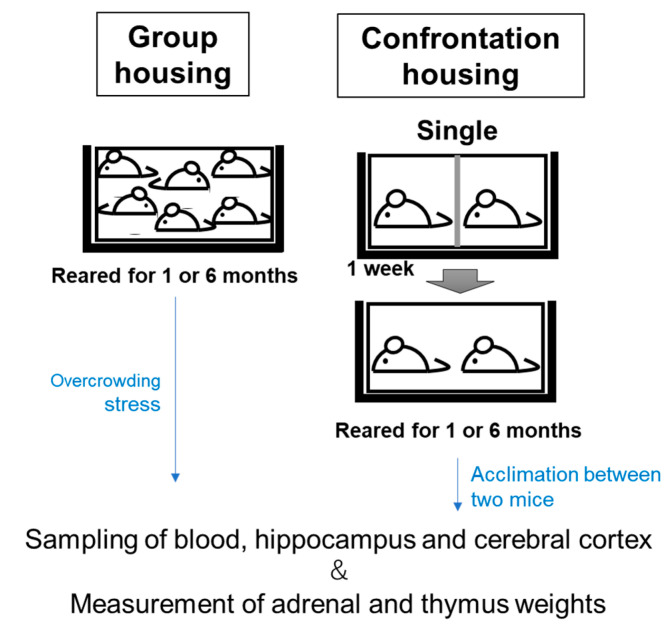
Experimental protocol for chronic stress loading.

**Figure 3 biology-12-00978-f003:**
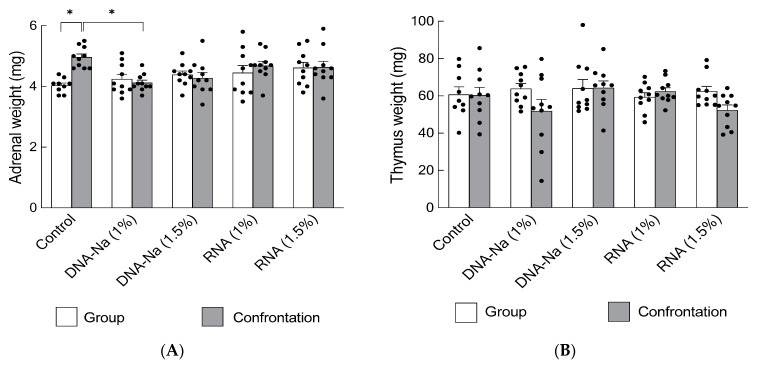
Effects of DNA-Na and RNA on adrenal glands (**A**) and the thymus (**B**) in stress-loaded mice. Each column bar represents the mean ± SEM (*n* = 10) overlaid on scatter plots (* *p* < 0.05, Tukey’s honestly significant difference method). Black dots indicate the values for each mouse.

**Figure 4 biology-12-00978-f004:**
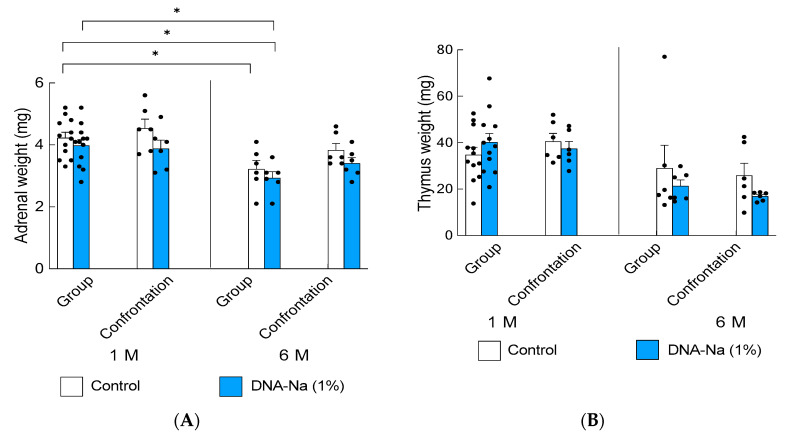
Effects of DNA-Na intake on adrenal glands (**A**) and thymus (**B**) over a longer period. Each column bar represents the mean ± SEM (1 M group, *n* = 12; others, *n* = 6) overlaid on scatter plots (* *p* < 0.05, Tukey’s honestly significant difference method). Black dots indicate the values for each mouse.

**Figure 5 biology-12-00978-f005:**
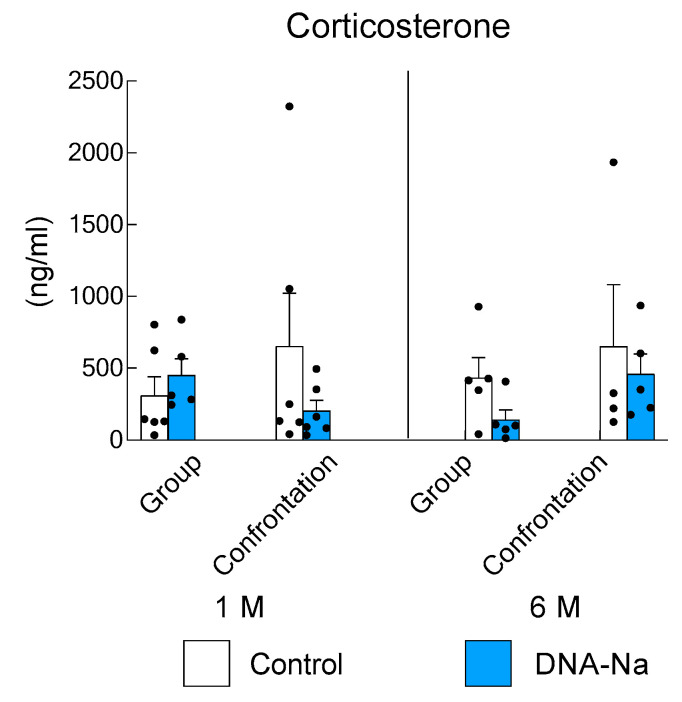
Effects of DNA-Na intake on the corticosterone levels. Each column bar represents the mean ± SEM (*n* = 6) overlaid on scatter plots. Black dots indicate the values for each mouse.

**Figure 6 biology-12-00978-f006:**
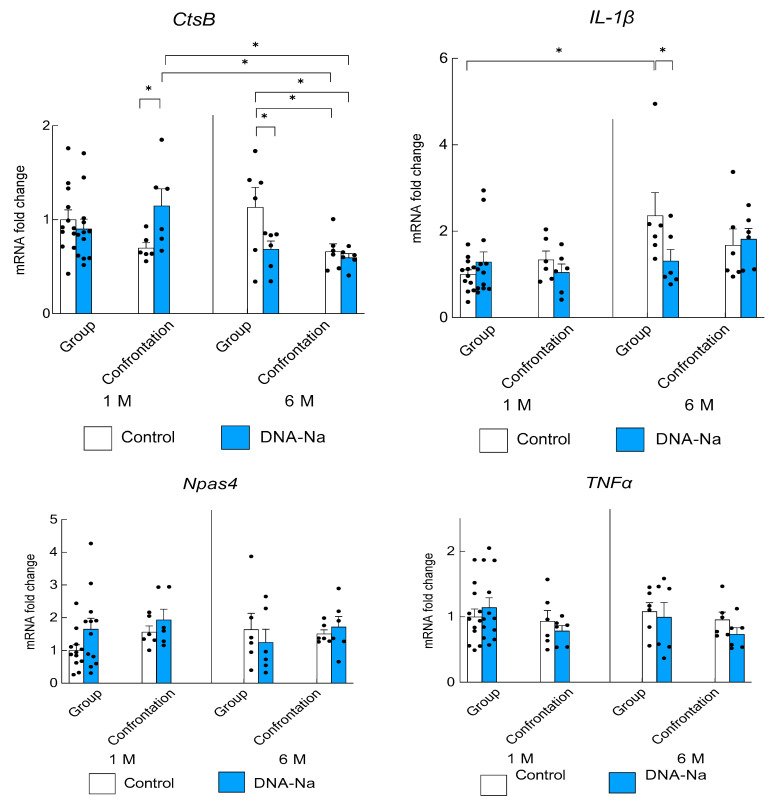
Effects of DNA-Na intake on the expression of *CtsB*, *IL-1β*, *Npas4*, and *TNFα* in the hippocampus. Each column bar represents the mean ± SEM (1 M group, *n* = 12; others, *n* = 6) overlaid on scatter plots (* *p* < 0.05, Tukey’s honestly significant difference method). Black dots indicate the values for each mouse.

**Figure 7 biology-12-00978-f007:**
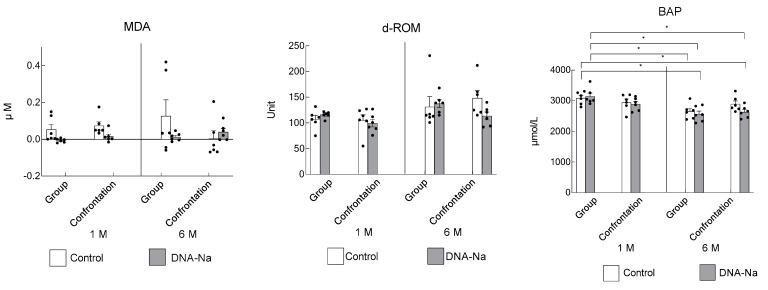
Effects of DNA-Na intake on oxidative damage. MDA was measured in the cerebral cortex. Both d-ROM and BAP were measured in the serum. Each column bar represents the mean ± SEM (*n* = 6) overlaid on scatter plots (* *p* < 0.05, Tukey’s honestly significant difference method). Black dots indicate the values for each mouse.

**Table 1 biology-12-00978-t001:** Sequences of the primers used in qRT-PCR.

Gene	Forward Sequence (5′ to 3′)	Reverse Sequence (5′ to 3′)	Ref.
*β-actin*	TGACAGGATGCAGAAGGAGA	GCTGGAAGGTGGACAGTGAG	
*Npas4*	AGCATTCCAGGCTCATCTGAA	GGCGAAGTAAGTCTTGGTAGGATT	[24]
*IL-1β*	GCAACTGTTCCTGAACTCAACT	ATCTTTTGGGGTCCGTCAACT	[25]
*TNFα*	CTGTCTACTGAACTTCGGGGTGAT	GGTCTGGGCCATAGAACTGATG	[26]
*CtsB*	CTGCTGAAGACCTGCTTA	AATTGTAGACTCCACCTGAA	[27]

## Data Availability

The data presented in this study are available upon request from the corresponding author.

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
