# Peer review of "Stress Reduction Potential in Mice Ingesting DNA from Salmon Milt"

_biology, 2023, doi:10.3390/biology12070978_

Round 1

Reviewer 1 Report

In their article “Stress Reduction in Mice Ingesting DNA from Salmon Milt” Keiko Unno and co-authors studied the functionality of food-derived nucleotide supplementation and its role in coping with the social behavioral and aging-related stresses on a murine model.

The role and functions of extracellular DNA as a biologically active molecule have been actively investigated in recent years. Taking into account the importance of investigations of aging and stress and the search for novel relieving factors, the relevance of the study is beyond doubt.

The study consists of two parts devoted to acute and chronic stress examination. In the first part, the authors revealed that alimentary DNA at 1%, but not RNA and not DNA at 1.5%, significantly reduced acute confrontational (social behavioral) stress in younger mice. The stress-characteristic adrenal hypertrophy was significantly suppressed in mice fed a diet containing 1% DNA-Na, while no significant changes in thymus weight were observed upon an exposure to alimentary DNA or RNA.

In the other series of tests, the authors explored chronic crowding stress in older mice. In this case, no effect of DNA-Na ingestion on adrenal (and thymus) gland size was observed. Studying the effect of DNA intake on serum levels of stress hormone corticosterone showed that DNA consumption tended to decrease them. Most interesting, an investigation of expression of the stress and inflammation-involved genes in the brain revealed that the expression of lysosomal CtsB, which is involved in induction of pro-inflammatory responses, was significantly higher in older group-rearing mice that ingested the control diet than in those that ingested DNA-Na. Similarly, the expression of pro-inflammatory cytokine IL-1β was significantly increased in older group mice fed the control diet, but significantly suppressed in older group-rearing mice fed DNA-Na. On the other hand, expression of TNFα, another pro-inflammatory cytokine, and Npas4, a factor important for stress tolerance, was not changed by DNA intake. Testing the oxidative damage in the cortex and serum antioxidant capacity also failed to reveal any difference between the groups with and without DNA-Na intake.

The authors concluded that ingestion of DNA derived from salmon milt suppresses age-related hippocampal inflammation induced by (chronic) crowding stress. I would recommend to add to the conclusion, that dietary DNA intake also reduces acute stress, as shown in confrontational murine tests.

The study results are correctly processed using the appropriate statistical methods. The conclusions are substantiated and may have heuristic significance for subsequent research in this field.

In my opinion, the study is worthy of being published in ‘Biology’ because the manuscript:

(a) reports interesting findings related to a comparatively novel stress-alleviating dietary factor,

(b) has qualitative signs of attractiveness for a wide variety of researchers (including, first of all, those who deal with the problems of aging, inflammation and stress signaling), and

(c) has a high citation potential.

So, I recommend accepting the manuscript with just a set of minor revisions suggested below.

The principle thing to improve greatly the manuscript is a shortage in the discussion. The authors have noted that further studies are needed to elucidate the detailed mechanisms of how DNA ingestion produces anti-inflammatory and anti-stress effects. But they did not mention an appreciable bulk of knowledge already obtained earlier on how the DNA fragments work as stress-signaling molecules (Ermakov et al. 2013; PMC3606786; Kostyuk et al. 2013; PMID: 23644378).

The reviewed study’s findings indeed well corroborate the notion of cell-free DNA as a hormetic stress-signaling factor (e.g. see Kostyuk et al. 2018; PMC5883976) especially when GC-rich (Malinovskaya et al. 2019; PMC6552851) and/or oxidized (Filev et al. 2019; PMC6644271). The manuscript contains information on nucleotide composition, but no data related to the length of DNA fragments in DNA-Na preparation applied. One can anticipate that the dietary preparation contained at least some fraction of GC-rich and/or oxidized oligonucleotide fragments, which is enough to activate DNA-receptors, first of all, TLR9 toll-like receptors and hence launch the inflammatory response. This action in turn triggers the anti-oxidant and anti-inflammatory homeostatic mechanisms and thus render an action reverse to the initial (i.e. are hormetic).

A good argument for this is the dosage fact reported. The authors unfortunately did not address at all the question, why DNA-Na had the anti-stress effect at 1%, but had no reliable effect at a higher concentration of 1.5%, whereas I deem it a key fact to prove the hormetic nature of the DNA action. The hormetic agents are known to act better at a low dose and for a short period of time (Schirrmacher, 2021; PMC8000639; Agathokleous&Calabrese, 2022; PMID: 35293720). The latter is in a good agreement with the result obtained in the study, that dietary DNA reduced adrenal gland size in acute stress, but not in chronic stress model – the fact which seems still not to be duly recognized by the authors.

It follows from the above-said, that when applied in practice as an anti-stress and anti-inflammatory agent, DNA should not be taken constantly, but by short courses and immediately after the beginning of or even before an anticipated stress periods (surgery, enhanced loads, winter periods, viral infections).

The other minor suggestions to improve the manuscript are the following.

Firstly, spelling. The text contains mistypes. For instance, in line 184: Aac-[cording]. I did not perform a total spell check, but I think more errors are not excluded. So I recommend to perform a total check. Spelling also requires unification. For instance, in line 327: proinflammatory, while in line 332: pro-inflammatory.

Secondly, abbreviations. Any abbreviation (except for commonly used like DNA or RNA) should be explained before used for the first time. The examples of unexplained abbreviations: d-ROM and BAP in line 191.

Finally, two minor report design suggestions are below.

I would like to advice to specify the exact p-value everywhere, whether the difference is deemed significant or not. Currently, the authors have specified only ‘p<0.05’ (when significant) or ‘tended’ (when insignificant).

Figure 4 seems redundant for me. The only information it gives is that older mice were heavier and mice that had been fed DNA were heavier than those that had not. It is not relevant to the article topic (stress-resistance and inflammation).

 PS. Waiver: all the references hereto are given for information only. I do not insist that the authors cite them, though I do not mind. Whether to cite or not to cite will be the authors’ decision. Any citation is advised only if the authors consider the report valuable and relevant.

Author Response

Response to Comments and Suggestions for Author 1

In their article “Stress Reduction in Mice Ingesting DNA from Salmon Milt” Keiko Unno and co-authors studied the functionality of food-derived nucleotide supplementation and its role in coping with the social behavioral and aging-related stresses on a murine model.

The role and functions of extracellular DNA as a biologically active molecule have been actively investigated in recent years. Taking into account the importance of investigations of aging and stress and the search for novel relieving factors, the relevance of the study is beyond doubt.

The study consists of two parts devoted to acute and chronic stress examination. In the first part, the authors revealed that alimentary DNA at 1%, but not RNA and not DNA at 1.5%, significantly reduced acute confrontational (social behavioral) stress in younger mice. The stress-characteristic adrenal hypertrophy was significantly suppressed in mice fed a diet containing 1% DNA-Na, while no significant changes in thymus weight were observed upon an exposure to alimentary DNA or RNA.

In the other series of tests, the authors explored chronic crowding stress in older mice. In this case, no effect of DNA-Na ingestion on adrenal (and thymus) gland size was observed. Studying the effect of DNA intake on serum levels of stress hormone corticosterone showed that DNA consumption tended to decrease them. Most interesting, an investigation of expression of the stress and inflammation-involved genes in the brain revealed that the expression of lysosomal CtsB, which is involved in induction of pro-inflammatory responses, was significantly higher in older group-rearing mice that ingested the control diet than in those that ingested DNA-Na. Similarly, the expression of pro-inflammatory cytokine IL-1β was significantly increased in older group mice fed the control diet, but significantly suppressed in older group-rearing mice fed DNA-Na. On the other hand, expression of TNFα, another pro-inflammatory cytokine, and Npas4, a factor important for stress tolerance, was not changed by DNA intake. Testing the oxidative damage in the cortex and serum antioxidant capacity also failed to reveal any difference between the groups with and without DNA-Na intake.

The authors concluded that ingestion of DNA derived from salmon milt suppresses age-related hippocampal inflammation induced by (chronic) crowding stress. I would recommend to add to the conclusion, that dietary DNA intake also reduces acute stress, as shown in confrontational murine tests.

The study results are correctly processed using the appropriate statistical methods. The conclusions are substantiated and may have heuristic significance for subsequent research in this field.

In my opinion, the study is worthy of being published in ‘Biology’ because the manuscript:

(a) reports interesting findings related to a comparatively novel stress-alleviating dietary factor,

(b) has qualitative signs of attractiveness for a wide variety of researchers (including, first of all, those who deal with the problems of aging, inflammation and stress signaling), and

(c) has a high citation potential.

So, I recommend accepting the manuscript with just a set of minor revisions suggested below.

The principle thing to improve greatly the manuscript is a shortage in the discussion. The authors have noted that further studies are needed to elucidate the detailed mechanisms of how DNA ingestion produces anti-inflammatory and anti-stress effects. But they did not mention an appreciable bulk of knowledge already obtained earlier on how the DNA fragments work as stress-signaling molecules (Ermakov et al. 2013; PMC3606786; Kostyuk et al. 2013; PMID: 23644378).

Thank you very much for reviewing our manuscript. We really appreciate you taking the time to share with us the papers on DNA fragments as stressors. We have not been able to successfully find such papers until now, and we are deeply grateful to you for sharing it with us. Based on this idea, we have rewritten the introduction and the entire discussion.

The reviewed study’s findings indeed well corroborate the notion of cell-free DNA as a hormetic stress-signaling factor (e.g. see Kostyuk et al. 2018; PMC5883976) especially when GC-rich (Malinovskaya et al. 2019; PMC6552851) and/or oxidized (Filev et al. 2019; PMC6644271). The manuscript contains information on nucleotide composition, but no data related to the length of DNA fragments in DNA-Na preparation applied. One can anticipate that the dietary preparation contained at least some fraction of GC-rich and/or oxidized oligonucleotide fragments, which is enough to activate DNA-receptors, first of all, TLR9 toll-like receptors and hence launch the inflammatory response. This action in turn triggers the anti-oxidant and anti-inflammatory homeostatic mechanisms and thus render an action reverse to the initial (i.e. are hormetic).

The length of the DNA fragments with respect to DNA-Na was not examined.

We would like to examine the length of the DNA fragments in the future, since the action of DNA may differ depending on the length of the DNA fragments.

Based on your suggestion, we immediately examined the expression of TLR9, a DNA receptor. However, in this experiment, we did not observe significant changes in TLR9 mRNA levels in the hippocampus of mice due to DNA-Na intake or stress loading (data not shown).

In the future, we would like to investigate DNA fragments detected in the plasma of mice subjected to long-term stress.

A good argument for this is the dosage fact reported. The authors unfortunately did not address at all the question, why DNA-Na had the anti-stress effect at 1%, but had no reliable effect at a higher concentration of 1.5%, whereas I deem it a key fact to prove the hormetic nature of the DNA action. The hormetic agents are known to act better at a low dose and for a short period of time (Schirrmacher, 2021; PMC8000639; Agathokleous&Calabrese, 2022; PMID: 35293720). The latter is in a good agreement with the result obtained in the study, that dietary DNA reduced adrenal gland size in acute stress, but not in chronic stress model – the fact which seems still not to be duly recognized by the authors.

We were having great difficulty understanding the lack of adrenal hypertrophy under long-term stress, but your explanation of the hormesis phenomenon, in which low doses and short periods of stress promote the stress response, while high doses and long periods of stress act in an inhibitory manner, has helped us to fully understand our experimental results. We revised the discussion. We deeply appreciate your meaningful suggestions.

The reasons why no dose-dependence was observed with 1% and 1.5% DNA-Na are also discussed in the discussion.

It follows from the above-said, that when applied in practice as an anti-stress and anti-inflammatory agent, DNA should not be taken constantly, but by short courses and immediately after the beginning of or even before an anticipated stress periods (surgery, enhanced loads, winter periods, viral infections).

Such use of DNA-Na supplements may be advisable. We added this in the discussion. (line 458-460)

The other minor suggestions to improve the manuscript are the following.

Firstly, spelling. The text contains mistypes. For instance, in line 184: Aac-[cording]. I did not perform a total spell check, but I think more errors are not excluded. So I recommend to perform a total check. Spelling also requires unification. For instance, in line 327: proinflammatory, while in line 332: pro-inflammatory.

Thanks. I have requested English editing again.

Secondly, abbreviations. Any abbreviation (except for commonly used like DNA or RNA) should be explained before used for the first time. The examples of unexplained abbreviations: d-ROM and BAP in line 191.

We revised them.

Finally, two minor report design suggestions are below.

I would like to advice to specify the exact p-value everywhere, whether the difference is deemed significant or not. Currently, the authors have specified only ‘p<0.05’ (when significant) or ‘tended’ (when insignificant).

The "trending" was revised to "no significant difference."

P-values were added. 

Figure 4 seems redundant for me. The only information it gives is that older mice were heavier and mice that had been fed DNA were heavier than those that had not. It is not relevant to the article topic (stress-resistance and inflammation).

Figure 4 was removed.

  1. Waiver: all the references hereto are given for information only. I do not insist that the authors cite them, though I do not mind. Whether to cite or not to cite will be the authors’ decision. Any citation is advised only if the authors consider the report valuable and relevant.

We have cited much of the references you provided. Thank you very much.

Reviewer 2 Report

This manuscript reports that addition of DNA sodium salt derived from salmon milt at a concentration of 1% to the food would reduce certain kinds of stress in mice, i.e., confrontation housing for one week and confrontation housing for 1 or 6 months.  Moreover, the expression of genes associated with hippocampal inflammation was said to be suppressed in the DNA-fed group.  Addition of RNA to the food had no such effect.  The authors conclude that dietary DNA intake may suppress inflammation in the brain caused by stress, which increases with age.  The work is generally well-described.  However, there are some issues with the data and the analysis.  Overall, the conclusions are insufficiently supported by the data, and thus the title is not acceptable and potentially misleading.  The authors could considerably tone down their interpretations, as this study revealed mostly negative data.

Major points

1.    The authors use the Fisher’s exact probability test and Tukey’s honestly significant difference test.  In almost all cases, they should run analysis of variance (ANOVA) tests, e.g., a 2-way ANOVA (housing condition x DNA/RNA added to the food), and if the ANOVA is positive, they should run a post-hoc test (e.g. Tukey’s).

2.    The authors find that the adrenal weight is increased with confrontation housing in control mice, but not in 1% DNA-fed mice.  It looks like that all other additions to the food (1.5% DNA-Na, 1% RNA and 1.5% RNA) also prevent the confrontation housing-induced increase in adrenal weight, but the authors write that “RNA had no significant effect”.

3.    Lines 255-256: “Thymus weights also tended to be lower in older than younger mice.” If there is no statistically significant difference, such a statement cannot be made.  This also applies to other places in the paper, e.g., lines 314, 381, 382.

4.    The chronic confrontation housing paradigm (6 months, Fig. 5) has the disadvantage that one cannot really say to which degree any observed effects are due to stress, to aging (and/or increase in body weight), or to an interaction between stress and aging.

5.    Overall, the data on the effects of feeding 1% DNA-Na on stress are not really convincing: reduction of adrenal weight (Fig. 3A), no difference in thymus weight (Fig. 3B) or body weight (Fig. 4), no difference after 1 months and 6 months of confrontation housing (Fig. 5) no difference in serum corticosterone levels (Fig. 6).

6.    The reported changes in mRNA expression are not more convincing: CtsB increased after confrontation housing (1M) but decreased after group housing (6M), IL-1beta decreased after group housing but not after confrontation housing ((6M), and no changes in Npas4 and TNFalpha expression were found.

7.    Also, in the measurements of oxidative damage (Fig. 8), there are no statistical differences with 1% DNA-Na in the food.

8.    Lines 484/485: “In the stress response, DNA-Na supplementation significantly suppressed the stress-induced increase in hippocampal inflammation rather than antioxidative damage.”  This sentence is not supported by the data.  The authors do not examine hippocampal inflammation.  They show an increase in mRNA concentration of a lysosomal enzyme and an interleukin that may be involved in inflammation, but Npas and the inflammatory maker TNFalpha are not changed.  There is no evidence presented that the increase in mRNA translates into an increased expression of proteins and that such an increase in protein expression would be functionally relevant.

9.    Line 495-497.  “Conclusions…Ingestion of DNA derived from salmon milt suppresses age-related hippocampal inflammation induced by crowding stress.”  For the reasons outlined, above, this concluding statement is not supported by the data.

Minor point

1.    Line 185: “on the Nikken Seil website”.  Please be more specific and provide the web address of the website that you are referring to.

Author Response

Response to Comments and Suggestions for Author 2

This manuscript reports that addition of DNA sodium salt derived from salmon milt at a concentration of 1% to the food would reduce certain kinds of stress in mice, i.e., confrontation housing for one week and confrontation housing for 1 or 6 months.  Moreover, the expression of genes associated with hippocampal inflammation was said to be suppressed in the DNA-fed group.  Addition of RNA to the food had no such effect.  The authors conclude that dietary DNA intake may suppress inflammation in the brain caused by stress, which increases with age.  The work is generally well-described.  However, there are some issues with the data and the analysis.  Overall, the conclusions are insufficiently supported by the data, and thus the title is not acceptable and potentially misleading.  The authors could considerably tone down their interpretations, as this study revealed mostly negative data.

Thank you for taking the time to review our manuscript. Due to your valuable comments, we have revised the title as follows: Stress reduction potential in mice fed salmon milt DNA. 

Major points

  1. The authors use the Fisher’s exact probability test and Tukey’s honestly significant difference test. In almost all cases, they should run analysis of variance (ANOVA) tests, e.g., a 2-way ANOVA (housing condition x DNA/RNA added to the food), and if the ANOVA is positive, they should run a post-hoc test (e.g. Tukey’s).

Thank you for your suggestion. I had overlooked it.

In fact, a one-way analysis of variance was used for the statistical analysis, with a statistical significance of p<0.05. The significance of the difference between the confidence interval and the mean was estimated using Tukey's honest significant difference method or Fisher's least significant difference test. This was added in Session 2.7.

  1. The authors find that the adrenal weight is increased with confrontation housing in control mice, but not in 1% DNA-fed mice. It looks like that all other additions to the food (1.5% DNA-Na, 1% RNA and 1.5% RNA) also prevent the confrontation housing-induced increase in adrenal weight, but the authors write that “RNA had no significant effect”.

Compared to the confrontation-reared control group, adrenal hypertrophy was lower in the DNA and RNA intake groups, as shown below, but statistically significant inhibition of adrenal hypertrophy was seen in the group that received the DNA-Na (1%) containing. The DNA-Na 1.5% group was a little more varied and unfortunately did not differ significantly by Tukey's test.

Rearing condition

DNA or RNA

Adrenal (mean ± SE) mg

Group

0%

4.03 ± 0.08

Confrontation

0%

4.96 ± 0.11

DNA-Na 1%

4.12 ± 0.09

DNA-Na 1.5%

4.28 ± 0.18

RNA 1%

4.68 ± 0.15

RNA 1.5%

4.63 ± 0.20

  1. Lines 255-256: “Thymus weights also tended to be lower in older than younger mice.” If there is no statistically significant difference, such a statement cannot be made. This also applies to other places in the paper, e.g., lines 314, 381, 382.

We revised them.

  1. The chronic confrontation housing paradigm (6 months, Fig. 5) has the disadvantage that one cannot really say to which degree any observed effects are due to stress, to aging (and/or increase in body weight), or to an interaction between stress and aging.

In the case of the ddY mice, the confrontation between the two mice almost disappears after long-term (more than 10 days) confrontation rearing, and the adrenal hypertrophy is no longer significant, suggesting that the mice under these conditions are hardly stressed. Therefore, we believe that the difference between 1 month and 6 months in confrontation rearing can be regarded mainly as an effect of aging.

On the other hand, in long-term rearing experiments, group rearing is a rearing condition that produces overcrowding stress. Therefore, we explained the effects of long-term rearing condition in Session 2.4 and Figure 2.

  1. Overall, the data on the effects of feeding 1% DNA-Na on stress are not really convincing: reduction of adrenal weight (Fig. 3A), no difference in thymus weight (Fig. 3B) or body weight (Fig. 4), no difference after 1 months and 6 months of confrontation housing (Fig. 5) no difference in serum corticosterone levels (Fig. 6).

Figure 4 was deleted because it only shows the increase in body weight with age.  

We had a very difficult time understanding why adrenal hypertrophy did not occur under long-term stress, but another reviewer introduced the hormesis phenomenon, in which low-dose, short-duration stress enhances the stress response, while high-dose, long-duration stress has an inhibitory effect, which helped us to fully understand our experimental results. We have completely rewritten the discussion section based on this idea.

  1. The reported changes in mRNA expression are not more convincing: CtsB increased after confrontation housing (1M) but decreased after group housing (6M), IL-1beta decreased after group housing but not after confrontation housing ((6M), and no changes in Npas4 and TNFalpha expression were found.

In 6-month group rearing mice under overcrowded stress, it is important to note that the expression of CtsB and IL-1β was significantly reduced by DNA-Na intake. On the other hand, the lack of a significant increase in expression in the 6-month confrontation rearing group also confirmed that the ddY mice were not stressed under the long-term confrontation rearing.

CtsB expression was increased by DNA-Na ingestion in mice in the 1-month confrontation rearing group. TNFα expression was unchanged. The reasons for these findings remain to be determined.

  1. Also, in the measurements of oxidative damage (Fig. 8), there are no statistical differences with 1% DNA-Na in the food.

Indeed, it was.

  1. Lines 484/485: “In the stress response, DNA-Na supplementation significantly suppressed the stress-induced increase in hippocampal inflammation rather than antioxidative damage.” This sentence is not supported by the data.  The authors do not examine hippocampal inflammation.  They show an increase in mRNA concentration of a lysosomal enzyme and an interleukin that may be involved in inflammation, but Npas and the inflammatory maker TNFalpha are not changed.  There is no evidence presented that the increase in mRNA translates into an increased expression of proteins and that such an increase in protein expression would be functionally relevant.

That was revised as follows: In the stress response, DNA-Na supplementation significantly suppressed increases in CtsB and IL-1β mRNA associated with stress-induced hippocampal inflammatory responses. (line 449-450)

  1. Line 495-497. “Conclusions…Ingestion of DNA derived from salmon milt suppresses age-related hippocampal inflammation induced by crowding stress.”  For the reasons outlined, above, this concluding statement is not supported by the data.

That was revised as follows: Ingestion of salmon milt-derived DNA suppressed the expression of hippocampal inflammation-related genes CtsB and IL-1β mRNA levels induced by crowding stress. (line 463-464)

Minor point

  1. Line 185: “on the Nikken Seil website”. Please be more specific and provide the web address of the website that you are referring to.

We added reference No.28.

Round 2

Reviewer 2 Report

The authors have addressed the points raised by this reviewer.